# Technical note: Apportionment of Southeast Asian Biomass Burning and Urban Influence via In Situ Trace Gas Enhancement Ratios

Joshua P. DiGangi<sup>1</sup>, Glenn S. Diskin<sup>1</sup>, Subin Yoon<sup>2,\*</sup>, Sergio L. Alvarez<sup>2</sup>, James H. Flynn<sup>2</sup>, Claire E. Robinson<sup>1,3,†</sup>, Michael A. Shook<sup>1</sup>, K. Lee Thornhill<sup>1,3</sup>, Edward L. Winstead<sup>1,3</sup>, Luke D. Ziemba<sup>1</sup>, Maria Obiminda L. Cambaliza<sup>4,5</sup>, James B. Simpas<sup>4,5</sup>, Miguel Ricardo A. Hilario<sup>6</sup>, Armin Sorooshian<sup>6</sup>

<sup>1</sup>NASA Langley Research Center, Hampton, VA, 23681, USA

Correspondence to: Joshua P. DiGangi (joshua.p.digangi@nasa.gov)

15 **Abstract.** Correlations in airborne in situ gas enhancement ratios of CH<sub>4</sub> to CO from the 2019 Cloud, Aerosol and Monsoon Processes Philippines Experiment (CAMP<sup>2</sup>Ex) field project over the Sulu, Philippine, and South China Seas were used to distinguish air masses with predominantly biomass burning, urban, or mixed influence, and identifying contributions from differing urban sources. Two approaches were created to produce a final data flag: one using a singular background for CO and CH<sub>4</sub> enhancement ratios and another determining enhancement ratios via linear regression of 4 min bins along the timeseries. HYSPLIT back trajectory analysis was used to identify air mass origins, and the resulting source regimes were examined for differences in ozone, reactive nitrogen, and aerosol chemical composition. ΔO<sub>3</sub>/ΔCO enhancement ratios were observed to be constant between urban source regimes, yet ΔNO<sub>3</sub>/ΔCO enhancement ratios differed across these regimes. For biomass burning sources, enhancement ratios in ΔO<sub>3</sub>/ΔCO were over a factor of two lower than those reported by previous studies in this region. Organic aerosol mass fractions were observed to be 2-3 times higher in biomass burning influenced regimes compared to urban regimes. This technique represents a simple yet powerful approach for separating emission influences in a chemically complex environment that enables identification and characterization of emission sources using exclusively routine measurements that can be accomplished with commonly available instrumentation.

### 1 Introduction

Biomass burning and urban fossil fuel emissions are both significant contributors to poor air quality. In particular, Southeast Asia is home to strong emissions of both types, and combined with many of the world's largest megacities, this contributes to widespread increased mortality due to respiratory diseases (Lee et al., 2018). Uncertainties in emissions and emission sources

<sup>&</sup>lt;sup>2</sup>University of Houston, Houston, TX, 77204, USA

<sup>&</sup>lt;sup>3</sup>Analytical Mechanics Associates, Inc., Hampton, VA, 23666, USA

<sup>&</sup>lt;sup>4</sup>Air Quality Dynamics Laboratory, Manila Observatory, Quezon City, 1101, Philippines

<sup>&</sup>lt;sup>5</sup>Department of Physics, Ateneo de Manila University, Quezon City, 1101, Philippines

<sup>&</sup>lt;sup>6</sup>University of Arizona, Tucson, AZ, 85721, USA

<sup>\*</sup>Now at: Scripps Institution of Oceanography, University of California San Diego, La Jolla, CA, 92037, USA †deceased

can exacerbate the difficulty in predicting these mortality effects (Marvin et al., 2024), thus improving the ability to accurately identify air mass emission sources, a crucial need to assist policymakers in crafting policies to reduce these premature deaths. Back trajectory models have been utilized to great effect in identifying physical air mass histories (Fleming et al., 2012; Hilario et al., 2021, 2024; Robinson et al., 2011) but are limited by the resolution and accuracy of the assimilated meteorological information driving the model which can poorly represent sub-grid scale emissions and overestimate long-range transport (Harris et al., 2005). Gas tracer enhancement ratios, which exploit differing emission ratios, can provide a chemical ground truth at the measurement site (Halliday et al., 2019; Smith et al., 2015). While not providing an air mass physical history, the enhancement ratio can provide insight into the chemical history of the air mass, making it an excellent complement to back trajectory models. The choice of gas tracers depends on the application and time scale of the emissions and subsequent transport. Gases must either have similar chemical lifetimes to be viable or have a stable, known emission profile so that their differing lifetimes can be used as a chemical clock (Barmet et al., 2012; Gelencsér et al., 1997). For evaluating larger-scale regional or mesoscale emissions, relative enhancements in more stable tracers have been used to great effect (Plant et al., 2022). Commonly, enhancement ratios of carbon monoxide (CO) to carbon dioxide (CO2) have been used to this end due to the direct relationship between their emission ratio and combustion efficiency (DiGangi et al., 2021; Halliday et al., 2019). However, this can be complicated in regions or seasons where biogenic or oceanic uptake of CO<sub>2</sub> can complicate the necessary stable tracer background (Silva et al., 2013).

In contrast, CO and methane (CH<sub>4</sub>) are inefficiently scavenged by precipitation and cloud processes due to their low solubility, leaving dominant photochemical loss pathways typically on the order of many days to weeks for CO (Holloway et al., 2000; Khalil and Rasmussen, 1984) and years for CH<sub>4</sub> (Cicerone and Oremland, 1988; Naik et al., 2013; Voulgarakis et al., 2013), slow on the scale of regional or mesoscale transport. Thus, correlated enhancements between CO and CH<sub>4</sub> (ΔCH<sub>4</sub>/ΔCO) can be primarily related to the air mass source. Biomass burning has been widely reported in the literature to have low ΔCH<sub>4</sub>/ΔCO enhancement ratios or emission ratios compared to the higher values reported from fossil fuel combustion. For example, based on shipborne measurements in the South China Sea while heavily influenced by biomass burning, Nara et al. (2017) reported a typical ΔCH<sub>4</sub>/ΔCO enhancement ratio of 7.95±0.21% ppb/ppb. Worden et al. (2017) report biomass burning ΔCH<sub>4</sub>/ΔCO emission ratios between 0-20% ppm/ppm over a variety of global domains based on Aura TES retrievals and GFED-driven GEOS-Chem modeling. In contrast, Helfter et al. (2016) observed CH<sub>4</sub>/CO flux emission ratios between 60-500% during urban eddy covariance flux studies over central London, UK, and Plant et al. (2019) observed mean ΔCH<sub>4</sub>/ΔCO enhancement ratios between 48-130% ppb/ppb from airborne studies over six Eastern US cities.

We present a new approach that combines multiple methods for calculating enhancement ratios of  $\Delta CH_4/\Delta CO$  to isolate contributions from differing chemical regimes observed during the 2019 Clouds, Aerosol and Monsoon Processes-Philippines Experiment (CAMP<sup>2</sup>Ex) field campaign (https://espo.nasa.gov/camp2ex). These chemical regimes were then compared with those determined through back trajectory analysis to corroborate the expected source regimes and predict their emission origin. Finally, the regimes were used to examine contributors to poor air quality (ozone, reactive nitrogen, aerosol) by attributed

source.

65

#### 2 Methods

#### 2.1 CAMP<sup>2</sup>Ex

CAMP<sup>2</sup>Ex was a joint field campaign sponsored by NASA, the Naval Research Laboratory, and the Manila Observatory in 2019 based in the Philippines (Reid et al., 2023), which predominantly sampled air masses over the Sulu, Philippine, and South China Seas. The overarching science questions were focused on improving understanding of the coupling of aerosol, cloud, and radiation processes in a complex chemical and meteorological environment. The measurement platforms included two aircraft, the NASA P-3 and the SPEC Learjet 35 based at Clark International Airport in Luzon, the Scripps R/V Sally Ride stationed in the Philippine Sea, and a ground station at Manila Observatory inside the Ateneo de Manila University in Quezon City, Metro Manila. Flight profiles ranged over the South China, Philippine, and Sulu Seas, capturing a variety of compositional and cloud environments (Hilario et al., 2021). The P-3 aircraft fielded a comprehensive payload, with a mixture of remote sensing and in situ cloud, aerosol, and composition instrumentation, data from which are the focus of this study. All flight data presented in this work are from P-3 in situ data averaged on a 5 s time base (i.e., a 5 s data merge), and all data are publicly available (NASA/LARC/SD/ASDC, 2020). These measurements sampled a variety of pollution sources during both the southwest monsoon and monsoon transition periods and provided an opportunity to examine how local and transported emissions affected air mass chemical composition and air quality.

# 2.2 Measurements

85

Gas phase CO2, CH4, and CO dry mole fractions were collected at 0.4 Hz resolution via a commercial near-infrared cavity ringdown spectrometer (G2401-m: Picarro, Inc.) fitted with a custom airborne sampling system to increase stability and reduce sampling artifacts (DiGangi et al., 2021). Two-point calibrations were performed hourly during flight using standards prepared by NOAA ESRL and traceable to the WMO X2007 (CO<sub>2</sub>), X2004A (CH<sub>4</sub>), and X2014A (CO) scales. Precisions for CO<sub>2</sub>, CH<sub>4</sub>, and CO during CAMP<sup>2</sup>Ex were 0.1 ppm, 1 ppb, and 5 ppb, and accuracies were 0.1 ppm, 1 ppb, and 2%, respectively. Ozone (O<sub>3</sub>) was measured using a pair of commercial dual-beam UV absorption sensors (Model 205: 2B Technologies, Inc.) fitted with a custom sampling manifold to enhance high altitude performance (Wei et al., 2021). Calibrations were performed in the laboratory prior to the campaign using a commercial NIST-traceable ozone generator (Model 306: 2B Technologies, Inc.). Due to an altitude-dependent discrepancy between the two sensors, final ozone data were reported as the average of the two measurements, with a combined uncertainty of 6% from both the intercomparison and the native uncertainty of the instrument. Water vapor mixing ratios were measured via open-path near infrared absorption spectroscopy by the NASA Diode Laser Hygrometer (Diskin et al., 2002). Nitrogen oxide (NO), summed total NO + nitrogen dioxide (NO<sub>2</sub>) (NO + NO<sub>2</sub> = NO<sub>3</sub>), and total reactive nitrogen (NO<sub>y</sub>) were measured using a two-channel chemiluminescence nitrogen oxide analyzer (Air Quality Design Inc.) sampling through an unheated PFA-sheathed inlet. NO and NO<sub>x</sub> were sampled serially on one channel using a 395 nm UV LED photolytic NO<sub>2</sub> converter (Air Quality Design Inc.). A molybdenum converter operating at 300°C mounted near the sample entry point to the aircraft cabin was used to convert NO<sub>v</sub> species to NO for detection. The NO, NO<sub>2</sub>, and NO<sub>v</sub>

limit of detection was 0.30, 0.27, 0.22 ppb with an uncertainty of 12%, 14%, and 12%, respectively. Submicron non-refractory aerosol mass concentrations were measured using a High-Resolution Time-of-Flight Aerosol Mass Spectrometer (HR-ToF-AMS, Aerodyne Research, Inc.) using standard data processing methods (Canagaratna et al., 2007; DeCarlo et al., 2006). HR-ToF-AMS measurements were obtained behind a pressure-controlled inlet for stability and were recorded in FastMS mode (25 s open, 5 s closed) and averaged to approximately 30 s for this analysis. Mass concentrations are reported at standard temperature and pressure (273 K and 1013.25 hPa), and sulfate, nitrate, ammonium, chlorine, and organic mass concentrations were determined using standard fragmentation tables (Allan et al., 2004). In this analysis, total AMS aerosol mass is defined as the sum of each of these five AMS species contributions.

# 3 Emission Flag Derivation

To separate out different chemical regimes observed in the CAMP<sup>2</sup>Ex dataset, specifically biomass burning versus urban emissions, this work leverages differences in the populations of enhancement ratios of CH<sub>4</sub> and CO. A combination of two different approaches to calculating  $\Delta$ CH<sub>4</sub>/ $\Delta$ CO enhancement ratios was used to isolate the biomass burning and urban regimes: one using a single campaign background to determine enhancement ratios (Sect. 3.1) and a second using dynamic linear fitting on a rolling time window throughout each flight to find correlated enhancement ratios (Sect. 3.2). The final flag merging these approaches was calculated for each timestamp in the 5 s data merge (Sect. 3.4).

# 3.1 Single Background Approach

The single background approach centers around the use of single background concentrations of CO and CH<sub>4</sub> for the purpose 115 of calculating mixing ratios. Conceptually, this approach assumes that emitted air is broadly mixing exclusively with some idealized regional background air. To determine these ideal background concentrations, subsets of data at high enhancement ratios were chosen that appeared to represent individual mixing lines and fit with a simple least squares linear regression. The most common intersection of these fits was chosen as the assigned background concentrations, which was 65 ppb for CO and 120 1.85 ppm for CH<sub>4</sub>. However, much of the data at small CH<sub>4</sub> and/or CO enhancements above these background levels were still not separable into urban and biomass burning regimes due to variability of the true background concentration of each species. Thus, it was necessary to find the minimum  $\Delta CH_4$  and  $\Delta CO$  enhancement above background at which the two regimes begin to be separable while also as low as possible to maximize the data capable of being assigned to each regime. To accomplish this, the frequency distribution of  $\Delta CH_4/\Delta CO$  enhancement ratios were examined at various  $\Delta CH_4$  and  $\Delta CO$  enhancement levels above background concentrations (Fig. 1a-b). In these plots, the high data frequency peaks of both the biomass burning 125 regime (at low  $\Delta CH_4/\Delta CO$  enhancement ratios) and the urban regime (at high  $\Delta CH_4/\Delta CO$  enhancement ratios) appear as high peaks off scale. As the cutoff enhancement concentrations were increased, a minimum in the frequency distribution emerges between 35-70% ppb/ppb ΔCH<sub>4</sub>/ΔCO (Fig. 1a-b, grey shaded area). The cutoff concentration enhancements at which this minimum emerges were determined to be the minimum enhancement at which the urban and BB data populations could be

isolated: 40 ppb for ΔCH<sub>4</sub> and 55 ppb for ΔCO. Once the cutoff concentration was reached in the CO or CH<sub>4</sub> time series, a deadband of an additional Δ10 ppb CO and CH<sub>4</sub> was added to avoid rapid regime switching influenced by instrument precision limitations, an example of which is shown in Figure S1. Flag categories were then assigned by enhancement ratio and intensity of enhancement with respect to these cutoff concentrations (Fig. 2a).

Background Biomass Burning 2.2 Mixing CH<sub>4</sub> (ppm) 2.1 2.0 No Correlation Biomass Burning 2.2 Low ΔCH<sub>4</sub>/ΔCO Urban High ΔCH<sub>4</sub>/ΔCO Urban CH4 (ppm) Urban/BB Mixing 2.1 Background High Ozone 2.2 Biomass Burning Urban/BB Mixing CH<sub>4</sub> (ppm) Urban No Corr 2.1 Low ∆CH<sub>4</sub>/∆CO Urban High ΔCH<sub>4</sub>/ΔCO Urban 2.0 200 400 600 800 CO (ppb)

Figure 1: Sensitivity test of  $\Delta CH_4/\Delta CO$  emission ratio distributions to concentration cutoffs in (a) CH<sub>4</sub> and (b) CO. Shaded area shows the minimum in  $\Delta CH_4/\Delta CO$  chosen to separate the regimes above the selected cutoffs at  $\Delta CH_4=40$  ppb and  $\Delta CO=55$  ppb. (c) Frequency distribution of  $\Delta CH_4/\Delta CO$  enhancement ratios showing rolling slope regimes, with dashed lines representing the slope cutoffs between regimes at 40% and 100%  $\Delta CH_4/\Delta CO$ .

Figure 2: Tracer correlation plot of CH<sub>4</sub> vs. CO mole fractions colored by regime excluding Clark-influenced data using the (a) single background method (black star denotes background values), (b) rolling slope method, (c) final combined method.

# 135 3.2 Rolling Slope Approach

As the single background technique is limited due to the coarse assumption of a constant chemical background over the entirety of the campaign, a different approach was used to extract more fine detailed information from the dataset. With this approach, the linear fits of CH<sub>4</sub> vs. CO were calculated using weighted orthogonal distance regression (ODRPACK95 - IGOR Pro v7;

Wu and Yu, 2018) for 4 min rolling windows through each flight, similar to the technique used by Halliday et al. (2019) and DiGangi et al. (2021) for  $\Delta$ CO/ $\Delta$ CO<sub>2</sub> enhancement ratios. Rolling windows with lower goodness of fit ( $r^2 < 0.5$ ) were filtered as uncorrelated. Slopes from the remaining fits represent a  $\Delta$ CH<sub>4</sub>/ $\Delta$ CO enhancement ratio product that is independent of changes in larger scale shifts in background concentration and more sensitive to mixing from recent emissions. In the frequency distribution of the  $\Delta$ CH<sub>4</sub>/ $\Delta$ CO slopes, three relative minima in the frequency distribution (Fig. 1c) were chosen to separate the dataset into four regimes (Table 1). One is a negative-slope regime that corresponded to mixing between urban-influenced and biomass burning-influenced air. A second regime was between 0-40% ppb/ppb  $\Delta$ CH<sub>4</sub>/ $\Delta$ CO, peaking near 10% ppb/ppb  $\Delta$ CH<sub>4</sub>/ $\Delta$ CO, with low slopes corresponding to biomass burning. The last two regimes had higher slopes consistent with urban influence: a moderate slope regime between 40-100% ppb/ppb  $\Delta$ CH<sub>4</sub>/ $\Delta$ CO, peaking around 60-80% ppb/ppb  $\Delta$ CH<sub>4</sub>/ $\Delta$ CO, and a higher sloped regime containing slopes above 100% ppb/ppb  $\Delta$ CH<sub>4</sub>/ $\Delta$ CO with a broad, multifeatured peak near 110-140% ppb/ppb and a long tail toward higher  $\Delta$ CH<sub>4</sub>/ $\Delta$ CO slopes. Finally, the resulting flag was manually filtered to merge clearly temporally contiguous air masses that may only have had correlation over a portion of the sampling region (Fig. 2b).

| Regime                                     | Single Background                                      |                                              | Rolling Slope                                 |
|--------------------------------------------|--------------------------------------------------------|----------------------------------------------|-----------------------------------------------|
|                                            | Enhancement Cutoff                                     | Enhancement Ratio Cutoff (ppb/ppb)           | ΔCH <sub>4</sub> /ΔCO Limits (ppb/ppb)        |
| Urban/Biomass<br>Burning Mixing            | $\Delta$ CO < 55 ppb $\Delta$ CH <sub>4</sub> < 40 ppb | -                                            | $\Delta \text{CH}_4/\Delta \text{CO} < 0\%$   |
| Biomass Burning                            | ΔCO > 55 ppb                                           | $\Delta \text{CH}_4/\Delta \text{CO} < 35\%$ | $0\% < \Delta CH_4/\Delta CO < 40\%$          |
| Urban Low<br>ΔCH <sub>4</sub> /ΔCO Regime  | $\Delta$ CO > 55 ppb $\Delta$ CH <sub>4</sub> > 40 ppb | 35% < ΔCH <sub>4</sub> /ΔCO < 70%            | $40\% < \Delta CH_4/\Delta CO < 100\%$        |
| Urban High<br>ΔCH <sub>4</sub> /ΔCO Regime | $\Delta \text{CH}_4 > 40 \text{ ppb}$                  | $\Delta \text{CH}_4/\Delta \text{CO} > 70\%$ | $\Delta \text{CH}_4/\Delta \text{CO} > 100\%$ |

Table 1: Cutoff concentration enhancements for CO and CH4 for both single and rolling slope background methods.

# 3.3 Special Cases

Data influenced by strong local sources during takeoff at Clark International Airport were flagged separately until the first sharp gradient in trace gas concentration/humidity or, if no gradient was discernible, until 1 km above ground level (AGL). Similarly, Clark influence prior to landing was flagged as either the last sharp trace gas concentration/humidity gradient or 1 km AGL through the end of the flight. A separate population of data was flagged as another special case shown in Fig. S2, where a lobe in the ΔCH<sub>4</sub>/ΔCO scatterplot (Fig. S2a) was identified due to a strong anticorrelation between ozone and water vapor data (Fig. S2b) at low water mixing ratios (< 8000 ppm<sub>v</sub>). This may be indicative of stratospheric mixing (Pan et al., 2014), which would normally also result in anti-correlations between ozone and both CH<sub>4</sub> and CO (Collins et al., 1993; Pan et al., 2004). In contrast, these ozone observations are uncorrelated with CO and appear positively correlated with CH<sub>4</sub> (Fig. S2c-d, respectively), analysis of which is deemed beyond the scope of this work. As these two cases are distinct from the rest of the dataset, both the Clark-influenced and potential stratospherically-influenced data were flagged as separate values in the data product and neglected in further analysis.

# 165 3.4 Final Combined Flag

The final flag background and biomass burning regimes were equivalent to those assigned by the single background method. The urban regime from the single background method was split into three separate regimes based on the correlated slope (or lack thereof) from the rolling slope method. The negative-slope correlated regime was combined with the urban/BB mixing regime from the single background method to form the final urban/biomass burning mixing regime. Figure 2c shows the final regime apportionments with respect to CH<sub>4</sub> and CO, while Table 1 shows the final cutoffs in both tracer enhancement and enhancement ratios for each method.

#### 4 Discussion

# 180 4.1 HYSPLIT Back Trajectory Comparison

To evaluate the utility of the emission flag and assign potential origins of the empirically-derived populations, hourly back trajectories were calculated using the NOAA Hybrid Single Particle Lagrangian Integrated Trajectory (HYSPLIT) model (Rolph et al., 2017; Stein et al., 2015), similar to those described in Hilario et al. (2021). Aircraft latitude, longitude, and altitude from the P-3 data merge were used to initialize the model for each 5 s point along the flight track for all flights. The Global Forecast System (GFS) 0.25° resolution reanalysis product from the National

Figure 3: Heat map of HYSPLIT 48 h back trajectories at 1 h resolution initialized every 5 s from flight track < 2 km, separated by regime: (a) background, (b) low ΔCH<sub>4</sub>/ΔCO urban, (c) high ΔCH<sub>4</sub>/ΔCO urban, (d) uncorrelated urban, and (e) biomass burning. (f) Heat map of VIIRS 375 m fire counts during September 2019.

Centers for Environmental Protection (NCEP) was used to drive the modeled meteorology. Figure 3 shows a summary of heat maps of the resulting hourly back trajectory points under 2 km above mean sea level, an altitude cutoff intended to isolate contributions to the back trajectories originating from the planetary boundary layer. Data were binned to 0.25° resolution where

the bin color denotes the frequency of a contribution. Each heat map represents back trajectories initialized during a different observed chemical regime.

Trajectories observed from the background chemical regime (Fig. 3a) were well dispersed around the region, though few back trajectories appear over land in the boundary layer within the 48 h time window. In contrast, trajectories observed during the biomass burning chemical regime (Fig. 3b) were focused toward the southwest of the flight domain. The bulk of these were focused from the direction of Borneo and Sulawesi, though contributions appear from as far as Sumatra, the Malay peninsula, and the rest of mainland Southeast Asia. These results were consistent with the larger clusters in September 2019 observed fire counts (Fig. 3f) from the VIIRS 375m product (NASA FIRMS, 2020). Of the correlated urban regimes, the low  $\Delta CH_4/\Delta CO$  urban regime back trajectories (Fig. 3c) were predominantly either from over continental China or from that direction. The high  $\Delta CH_4/\Delta CO$  urban regime was observed more frequently and appears from the back trajectories (Fig. 3d) to be focused locally in and around Luzon. The most frequent contributions in the uncorrelated urban regime (Fig. 3e) seemed to originate from Korea and Kyushu (Japan), the distance from which could be an explanation for existence of consistent CO and CH<sub>4</sub> enhancements with no short-term correlations between the species. The partitioning of the back trajectories was based only on the  $\Delta CH_4/\Delta CO$ -derived chemical regimes, yet results in distinct separation in the back trajectory regimes, providing evidence toward the validity of this technique.

Figure 4: Tracer-tracer plots of (a) O<sub>3</sub> and (b) NO<sub>y</sub> versus CO separated by chemical regime. Trend lines denote linear fits for their respective regimes.

Figure 5: Relative aerosol chemical composition with respect to chemical regime: (a) background, (b) biomass burning, (c) biomass burning/urban mixing, (d) uncorrelated urban, (e) low  $\Delta CH_4/\Delta CO$  urban, (f) high  $\Delta CH_4/\Delta CO$  urban. (g) Average total AMS mass concentration by regime.

# 4.2 Ozone and Reactive Nitrogen Regime Dependencies

The ozone and reactive nitrogen measurements can give insight into the relative contributions within the different chemical regimes observed during the campaign. Figure 4a shows enhancements in ozone with respect to CO for each regime. Biomass burning ΔO<sub>3</sub>/ΔCO is very strongly correlated (r² 0.93) with a slope of 7.6±0.3% ppbv/ppb, which is lower than both the 18±8% ppbv/ppb reported by Lin et al. (2013) and the 20±1% ppbv/ppbv reported by Kondo et al. (2004) in roughly the same region. Seasonality may affect these results somewhat, as the Lin et al. (2013) measurement values represent an annual range and the Kondo et al. (2004) values were only during February-April, both contrasting with the measurements exclusively during August-October in this work. Otherwise, this may be indicative of a change in emission patterns over the previous decade, with lower O<sub>3</sub> production implying different NO<sub>x</sub> or VOC loading. Urban ΔO<sub>3</sub>/ΔCO enhancements from all categories were not readily separable, but taken as one, they were still well correlated (r² 0.71) with a slope of 35±2% ppbv/ppb.

This consistency in  $\Delta O_3/\Delta CO$  correlations among the urban regimes did not persist for the reactive nitrogen/CO ratios.  $\Delta NO_y/\Delta CO$  enhancements were used to represent reactive nitrogen enhancements in the different regimes, as shown in Fig. 4b. Biomass burning  $\Delta NO_y/\Delta CO$  was not as correlated as  $\Delta O_3/\Delta CO$  with an  $r^2$  of 0.45 and a slope of 0.144±0.002 pptv/ppb.  $\Delta NO_y/\Delta CO$  enhancements were similarly correlated in the high and low  $\Delta CH_4/\Delta CO$  urban regimes with  $r^2$  values of 0.33 and 0.38, respectively. However, in contrast with the  $\Delta O_3/\Delta CO$  urban correlations, the high and low  $\Delta CH_4/\Delta CO$  urban regimes exhibited distinctive  $\Delta NO_y/\Delta CO$  slopes, with the high  $\Delta CH_4/\Delta CO$  regime slope (8.7±0.2 pptv/ppb) a factor of six greater than the low  $\Delta CH_4/\Delta CO$  urban regime slope (1.38±0.02 pptv/ppb). This range of ratios is comparable to those reported by Kondo

et al. (2004), who observed  $\Delta NO_y/\Delta CO$  slopes of 2.2±0.6 pptv/ppbv in the planetary boundary layer and 7.9±0.3 pptv/ppbv in the lower troposphere over Southeast Asia in primarily urban-influenced air during TRACE-P in 2001. In general, higher NO<sub>y</sub>/CO emission factors are indicative of higher efficiency combustion, as high temperatures lead to both more complete conversion of carbon to  $CO_2$  and greater  $NO_x$  production. This would infer that urban combustion sources sampled in the high  $\Delta CH_4/\Delta CO$  regime were on average more efficient than those sampled in the low  $\Delta CH_4/\Delta CO$  regime. More broadly, the common ozone dependency under these drastically different reactive nitrogen environments suggests that ozone production in the low and high  $\Delta CH_4/\Delta CO$  urban regimes may have been VOC-limited.

Figure 6: Relative AMS mass fractions of (a) organics and (b) sulfates versus total AMS mass concentration by chemical regime.

#### 4.3 Aerosol Composition Regime Dependencies

To evaluate the chemical regime effects on aerosol composition, Fig. 5a-f shows the relative speciated contribution to aerosol mass observed by the AMS in each regime integrated over the entire campaign, while Fig. 5g shows the total aerosol mass observed in each regime, also over the entire campaign. The biomass burning regime (Fig. 5b) exhibited the highest average aerosol mass concentration ( $13 \mu g/m^3$ ) and the largest relative mass contribution from organic aerosols (77%). The BB/urban mixing regime (Fig. 5c) exhibited nearly identical relative aerosol contributions to the biomass burning regime, likely related to the observed larger mass loading from biomass burning compared to urban emissions. Among the urban regimes, the highest average mass concentration was observed in the low  $\Delta CH_4/\Delta CO$  urban regime ( $11 \mu g/m^3$ ), with relative aerosol contributions very similar to those observed in the background regime: ~40% each organic and sulfate aerosol mass. The high  $\Delta CH_4/\Delta CO$  urban regime exhibited less aerosol on average ( $8.8 \mu g/m^3$ ) than the low  $\Delta CH_4/\Delta CO$  urban regime ( $10.9 \mu g/m^3$ ) and had very similar composition with also ~40% each organic and sulfate aerosol mass. In the background regime (Fig. 5a), average aerosol loading was much lower (~ $0.7 \mu g/m^3$ ) than for other regimes, and the largest relative contribution was from sulfate aerosols, with 49% of the loading, followed by organic aerosols for another 39%. In contrast, the biomass burning (Fig. 5b) and BB/urban (Fig. 5c) mixing regimes had similar relative mass contributions with respectively 77% and 76% of the observed aerosol mass contribution from organic aerosol and only about 14% and 15% respectively from sulfate aerosol. Between the

urban regimes, the uncorrelated regime had the lowest average aerosol mass but a larger relative sulfate aerosol contribution (62%) compared to the low and high  $\Delta CH_4/\Delta CO$  urban regimes (44% and 37% respectively).

Figure 6 shows the trend in relative mass fraction contributions for the two dominant aerosol mass contributors, organic and sulfate, as a function of total aerosol mass in the biomass burning and urban chemical regimes. Within the urban regimes, there was large variability in the mass fractions for both organic and sulfate. One exception seems to be that the mass fractions asymptote at nearly 50% (organic) and 35% (sulfate) of total aerosol mass within the low  $\Delta$ CH<sub>4</sub>/ $\Delta$ CO urban regime, though this could be an example of the more limited observations of this regime. In the biomass burning regime, the mass fractions have high total mass asymptotes near 80% (organic) and 10% (sulfate).

# **5 Summary**

Enhancement ratios in airborne in situ CH<sub>4</sub> and CO were used to isolate predominant sources of air mass emission influence during the CAMP<sup>2</sup>Ex campaign over the western Pacific near the Philippines. Due to a more variable background in CO<sub>2</sub> during CAMP<sup>2</sup>Ex,  $\Delta$ CH<sub>4</sub>/ $\Delta$ CO was observed to exhibit greater utility in discerning combustion emission sources in comparison to the more commonly used  $\Delta$ CO/ $\Delta$ CO<sub>2</sub>, a metric of combustion efficiency. A 5 s data product was derived for all flights assigning regimes of biomass burning and various urban sources based on natural groupings of ranges of  $\Delta$ CH<sub>4</sub>/ $\Delta$ CO enhancement ratios. HYSPLIT back trajectory modeling was used to predict the likely emission origins of the differing regimes and to corroborate the method, with a higher  $\Delta$ CH<sub>4</sub>/ $\Delta$ CO corresponding to local emissions and a lower  $\Delta$ CH<sub>4</sub>/ $\Delta$ CO corresponding to continental outflow.  $\Delta$ O<sub>3</sub>/ $\Delta$ CO were individually strongly correlated in both biomass burning and urban regimes, but with a much lower slope for biomass burning (7.6±0.3% ppbv/ppb) compared to the slope for combined urban regimes (35±2% ppbv/ppb).  $\Delta$ NO<sub>y</sub>/ $\Delta$ CO were also highly correlated but had a sixfold higher slope for the local high  $\Delta$ CH<sub>4</sub>/ $\Delta$ CO urban regime compared to the continental outflow low  $\Delta$ CH<sub>4</sub>/ $\Delta$ CO urban regime. Organic aerosols were observed to dominate aerosol mass fractions for biomass burning influenced regimes at ~75% of the total mass fraction, whereas background as well as low and high  $\Delta$ CH<sub>4</sub>/ $\Delta$ CO urban regimes were more evenly balanced between sulfate and organic aerosols.

The multi-faceted method developed in this analysis provides a novel approach to assessing regional source contributions. One caveat is that this method appears to work well at receptor sites away from direct sources as in CAMP<sup>2</sup>Ex but may not work as well very close to emission sources or in air masses simultaneously influenced by both biomass burning and fossil fuel combustion. The use of these simple tracer relationships perhaps achieved using common, commercially-available instrumentation, may provide a feasible approach for constraining certain emission sources without requiring the cost and complexity of a larger-scale experimental effort. Thus, this method can more broadly assist scientists and policymakers in understanding the impact of emission type on air quality metrics, particularly in regions where data from more complex instrumentation (e.g. PTR-MS, GC) are sparse.

# **Data Availability**

All CAMP<sup>2</sup>Ex airborne data is available from NASA on the project data archive at https://doi.org/10.5067/SUBORBITAL/CAMP2EX2018/DATA001 (NASA/LARC/SD/ASDC, 2020). The HYSPLIT model is a product of NOAA and available at https://www.ready.noaa.gov/HYSPLIT.php. VIIRS fire count data is available from NASA at https://firms.modaps.eosdis.nasa.gov/download/ (NASA FIRMS, 2020).

#### **Author Contributions**

JPD, GSD, SY, SLA, JHF, CER, MAS, KLT, ELW, and LDZ participated in the data collection. JPD, GSD, LDZ, MOLC, JBS, MRAH, and AS performed the data analysis and provided feedback. JPD prepared the manuscript with contributions from all coauthors.

# 300 Competing Interests

AS is an editor of Atmospheric Chemistry and Physics.

# Acknowledgements

The authors thank NASA, the Naval Research Laboratory, and Manila Observatory for their support. Thanks to Mario Rana of Analytical Mechanics Associates, Inc. as well as Jim Plant and Jimmy Geiger from NASA for their technical assistance.

The authors gratefully acknowledge the NOAA Air Resources Laboratory (ARL) for the provision of the HYSPLIT transport and dispersion model and READY website (https://www.ready.noaa.gov) used in this publication. We acknowledge the use of data and/or imagery from NASA's Fire Information for Resource Management System (FIRMS) (https://earthdata.nasa.gov/firms), part of NASA's Earth Observing System Data and Information System (EOSDIS). University of Arizona investigators were supported by NASA grant 80NSSC18K0148 in support of CAMP<sup>2</sup>Ex.

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
