# Peer review of "Technical note: Apportionment of Southeast Asian Biomass Burning and Urban Influence via In Situ Trace Gas Enhancement Ratios"

_EGUsphere, 2025_

## Author Comment (AC1)

Review of "Technical note: Apportionment of Southeast Asian Biomass Burning and Urban Influence via In Situ Trace Gas Enhancement Ratios" by DiGangi and coauthors

Author responses in red. We thank the reviewer for their time and detail examining our manuscript!

This is a straightforward technical note presenting the usefulness of enhancement ratios of CH4 to CO to identify/constrain various origin source signatures in regions influenced by various air masses transported to a measurement location. This technique is reasonably sound and appears to be useful for apportioning data from different source regimes in the absence of measurements of more specific chemical tracers (i.e., VOCs or other non-organic gases like HCN that are emitted nearly exclusively from specific anthropogenic or biomass burning emissions sources.) This should be made clear in the paper.

We have added language to both the conclusion and abstract to emphasize this point.

Once this and the following comments and technical corrections are addressed, this technical note should be published in ACP.

Lines 129-130 and Figure S1: The brief description of the use of a ±5 ppb CO and CH4 hysteresis could use a little more explanation. The dashed lines for CH4, 1.85 + 0.04 = 1.89 ppm, and ± 5 ppb (± 0.005 ppb) would be 1.885 – 1.895, and for CO, 65 ppb + 55 ppb = 120 ppb, and then ± 5 ppb would be from 115-125 ppb? The dashed lines in Fig. S1 are each 5 ppb above those (1.89 – 1.90 ppm for CH4 and 120 – 130 for CO), which seem too high for the explanation given in the text.

Indeed, this should have been described as an additional Δ10 ppb hysteresis, and the figure as it exists in the supplement is the correct representation. We have altered the text to correct this error.

Technical corrections:

Lines 14, 73, 78, etc.: "Seas" should be capitalized.

We have corrected this in all instances.

Lines 21, 22, 23, 36, 38, etc.: "air mass" and "air masses" should both be two words.

We have corrected this in all instances.

Line 29: I believe it should be "enables" (novel approach is singular).

We have made this correction.

Line 53: it would be better to spell out "many days to weeks".

We have made this correction.

Lines 118, 137, and Figs. 2 and S2 captions: "vs." should have a period.

We have made this correction.

Line 130: Supplemental Fig. S1. (Technically, "Supplemental" isn't needed, either – the S is sufficient.) (Similarly, Fig. S2 – line 157).

We have made this correction.

Figure 2 caption: I recommend making this a proper sentence: "… colored by regime excluding Clark-influenced data using (a) a global background method, (b) a rolling slope method, and (c) a final combined method."

We have made this correction.

Table 1: in the first column there are two $CH_4$s that need the 4s subscripted.

We have made this correction.

Lines 156-158: This sentence seems awkward. I recommend either add another comma, or change the comma to a semicolon and add a "was" before identified, maybe?

The sentence was clarified to now read; "A separate population of data was flagged as another special case shown in Fig. S2, where a lobe in the $\Delta CH4/\Delta CO$ scatterplot (Fig. S2a) was identified due to a strong anticorrelation between ozone and water vapor data (Fig. S2b) at low water mixing ratios (< 8000 ppmv)."

Line 160: remove " Jr." -- generally, suffixes aren't included in in-text citations.

We have made this correction.

Line 161: Fig. S2c-d -- it is still a single figure being referenced.

We have made this correction.

Lines 182, 191, Fig. 3 caption, etc.: "back trajectory" and "back trajectories" should each be two words.

We have made this correction.

Line 191: "Figure 3 shows…"

We have made this correction.

Line 219: "Figure 4a shows…"

We have made this correction.

Line 223: remove "the" before "February-April"

We have made this correction.

Line 228: "Fig. 4b"

We have made this correction.

Line 215: The legend colors in Fig. 5 do not correspond to the colors in the pie charts in (a)-(f).

We have corrected the ammonium color in the legend.

Lines 245 and 253: Maybe use "BB/urban" similar to the Fig. 5g category name instead of "biomass/urban", to be clear that this isn't a mixture of biogenic and urban emissions.

We have made this correction.

Line 250: "Fig. 5a"

We have made this correction.

Line 257: "Figure 6 shows…"

We have made this correction.

Line 260: delete one "urban".

We have made this correction.

Line 270: remove "sloped" x2: "… with higher $\Delta CH_4/\Delta CO$ corresponding to local emissions and lower $\Delta CH_4/\Delta CO$ corresponding…" Similarly, consider using "relationship" instead of "slope" in the rest of this paragraph.

We have made this correction.

Lines 305-end: I believe the journal names should be abbreviated.

We have corrected the references to include abbreviated journal names.

Line 347: there is a rogue "$" in the CAMP$^2$Ex name.

We have made this correction.

Line 411: CO$_2$ should have a subscripted 2.

We have made this correction.

---

## Author Comment (AC2)

This paper presents a new method that uses the gas enhancement ratios of CH4 to CO for apportionment of different airmass sources during the CAMP2Ex campaign. The developed method is very useful and the results presented are also very interesting. Below are a few minor comments/suggestions.

We thank the reviewer for their time and attention reviewing the manuscript!

Is "Technical note:" necessary in the title?

Our understanding is that this is a requirement for the publication type, as was submitted as a technical note. We defer to the editor and editorial staff's judgement.

The abstract is somewhat difficult to follow, as it takes several sentences before clearly stating the main objective of the study—using $CH_4$-to-CO enhancement ratios to separate airmass influences. The initial portion of the abstract, while informative, may distract from the core contribution and could be shortened or moved to the introduction. I suggest streamlining the abstract to more quickly convey what was done and what was found in this study.

We have reorganized and edited the abstract, removing some of the earlier information about the field campaign, some of which has been moved to Section 2.1.

Can you provide a figure showing the flight tracks of CAMP2Ex?

A map has been added as Fig. S1 with flight tracks colored by altitude AGL.

I suggest providing more details and explanations in section 3.1 and 3.2, as the methods described are not very straightforward to follow.

We have added more explanation to especially the beginning of Sect. 3.1, which we believe makes the overall description more clear.

Figure 3, what is "BT"?

BT stands for back trajectory, and the figure has been altered to state this fully.

Line 231: "the high and low $\Delta CH4/\Delta CO$ urban regimes exhibited distinctive $\Delta NOy/\Delta CO$ slopes". This is a particularly interesting result. I encourage the authors to elaborate further on the potential reasons behind these differences and their implications for understanding urban emission sources or atmospheric processing.

We have added the following text toward the end of that paragraph: "In general, higher $NO_y$/CO emission factors are indicative of higher efficiency combustion, as high temperatures lead to more complete conversion of carbon to $CO_2$ and greater $NO_x$ production. This would

infer that urban combustion sources sampled in the high $\Delta CH_4/\Delta CO$ regime were on average more efficient that those sampled in the low $\Delta CH_4/\Delta CO$ regime."

Line 249: "The high $\Delta CH4/\Delta CO$ urban regime exhibited less aerosol on average (8.8 µg/m3) than the low $\Delta CH4/\Delta CO$ urban regime and had very similar composition with also ~40% each organic and sulfate aerosol mass" This is interesting. Just out of curiosity, any possible explanations?

Unfortunately, the project dataset does not include any further composition measurements that would allow for us to discern more subtle differences (differing organic compound compositions, for example), making speculation difficult. It could indicate a similar level of sulfur impurities in the respective fossil fuel sources, which could lead to the similarity in ratios. Without some other method of substantiation, we felt that this level of speculation seemed outside of the scope of the manuscript.